# Prevalence of non-communicable diseases among individuals with HIV infection by antiretroviral therapy status in Dar es Salaam, Tanzania

Irene Kato[ID]‡*, Basil Tumaini[ID]‡, Kisali Pallangyo

Department of Internal Medicine, Muhimbili University of Health and Allied Sciences, Dar es Salaam, Tanzania

‡ These authors share first authorship on this work.
* drirene86@gmail.com

## Abstract

### Background

Long-term antiretroviral therapy has modified the clinical course of HIV infection to a chronic condition associated with increased risk of developing non-communicable diseases (NCDs). Information is scant, from sub-Saharan Africa, on the prevalence of NCDs and associated factors among individuals on ART.

### Methodology

We consecutively enrolled individuals with HIV infection who were ART naïve and those on ART for ≥5 years (LTART) attending health facilities in Dar es Salaam. Participant's blood pressure, anthropometric measurements, and fasting blood glucose were recorded. Participants with impaired fasting blood glucose underwent an oral glucose tolerance test. A venous blood sample was sent to the lab for biochemical tests. Chi-square test was used to compare proportions, Poisson regression with robust standard errors was used to determine associations between variables.

### Results

Overall, 612 individuals with HIV infection were enrolled, half of whom were ART naïve. Females comprised 71.9% and 68.0% of participants in the LTART and ART naïve study arms, respectively, p = 0.290. The mean age (±SD) was 44.9 ± 12.7 years and 37.5 ± 11.8 years among LTART and ART naïve participants, respectively, p<0.001. Hypertension was documented in 25.2% in those on LTART compared to 6.9% among ART naïve subjects, p<0.001. Impaired glucose tolerance was found in 22.9% and 4.6% among LTART compared to ART naïve subjects, p<0.001. Diabetes mellitus was detected in 17.0% of those on LTART compared to 3.9% ART naïve participants, p<0.001. Hypercholesterolemia was found in 30.4% of individuals on LTART compared to 16.7% of ART naïve subjects, p<0.001, and hypertriglyceridemia was found in 16.0% of participants on LTART compared

**Data Availability Statement:** All relevant data are uploaded to the OSF database and publicly accessible via the following URL: https://osf.io/

vczy9/?view_only=
fa6643e3f077493d8f28a584b30b1fe0.

**Funding:** The author(s) received no specific funding for this work.

**Competing interests:** The authors have declared that no competing interests exist.

to 9.5% of ART naïve, p = 0.015. LTART use, age $\geq$40 years, history of smoking, and body mass index were independently associated with NCDs.

## Conclusion

Hypertension, impaired glucose tolerance, diabetes mellitus, hypercholesterolemia, and hypertriglyceridemia were associated with long-term use of antiretroviral drugs.

## Introduction

HIV has affected parts of the world variably with sub-Saharan Africa having about two-thirds of the total number of people living with HIV (PLWHIV). The introduction of antiretroviral therapy has changed the natural history of HIV infection with a decrease in mortality due to AIDS-related illnesses, opportunistic infections and malignancies [1].

HIV disease and antiretroviral therapy (ART) have been shown to increase the risk of metabolic syndrome predisposing to type 2 diabetes mellitus, cardiovascular and renal diseases [2]. Consequently, in addition to the usual risk factors for non-communicable diseases (NCDs) seen in the general population, PLWHIV may have additional risks. Endothelial dysfunction, as well as metabolic disorders associated with HIV-related chronic inflammation and the use of antiretroviral drugs that cause toxicity through direct or indirect effects, may be responsible for the observed excess risk of NCDs [2,3].

With over 35 million people living and aging with HIV, a new global challenge of addressing morbidity and mortality due to NCDs in PLWHIV is looming [4]. NCDs tend to increase with age and are prevalent in PLWHIV where HIV disease and its treatment are being implicated in the causation. The use of effective ART has led to significant increases in the survival and quality of life for people with HIV giving an average life expectancy increase of approximately 13 years in the western countries [5].

Information is scant, from sub-Saharan Africa, on the prevalence of NCDs among individuals on long-term ART including their risk factor profile. Such information is vital to inform clinicians, hospital managers and policymakers in the provision of optimum care for individuals with HIV infection as well as maintain the gains already made in the fight against HIV.

This study aimed at determining the prevalence of selected NCDs and associated factors among individuals with HIV infection on long-term ART ($\geq$5 years) compared to ART naïve subjects receiving health care facility services in Dar es Salaam.

## Materials and methods

### Ethics statement

Ethical approval for the study was obtained from the Research and Publication Committee of Muhimbili University of Health and Allied Sciences. Permission to conduct the study was obtained from Muhimbili National Hospital administration as well as Kinondoni and Temeke Municipal councils. Written informed consent was obtained from all participants before enrolment. The confidentiality of patient information was ensured.

### Study design and population

This was an analytical cross-sectional study among individuals with HIV infection in Dar es Salaam, a city with a population of over 4 million people. It was conducted from September to

December 2017, at HIV care and treatment clinics (CTC) and provider-initiated testing and counseling (PITC) rooms of 4 centers: Muhimbili National Hospital–a tertiary referral hospital; Mwananyamala hospital and Temeke hospital–regional referral hospitals; and Mbagala hospital–a district hospital. Each of the four sites serves 60–200 individuals with HIV infection daily, irrespective of their specific geographical residence within the Dar es Salaam region. We consecutively enrolled registered individuals with HIV infection aged ≥18 years, who were either newly diagnosed and ART naïve or had been on ART for ≥5 years. Individuals with HIV infection who were pregnant were excluded.

## Sample size estimation

To determine the minimum sample size required, we used the formula for comparing two proportions, $n = (Z_{\alpha/2}+Z_\beta)^2*(p_1(1-p_1) + p_2(1-p_2))/(p_1-p_2)^2$, where, $Z_{\alpha/2}$ is the critical value of the Normal distribution at $\alpha/2$ (for a confidence level of 95%, $\alpha$ is 0.05 and the critical value is 1.96), $Z_\beta$ is the critical value of the Normal distribution at $\beta$ (for a power of 80%, $\beta$ is 0.2 and the critical value is 0.84) and $p_1$ and $p_2$ are the expected sample proportions of the two groups or proportions observed in a similar study. We used $p_1$ = 32% and $p_2$ = 22%, basing the estimates on the prevalence of hypertension in people with HIV infection on ART versus ART naïve, respectively, in a study conducted in Dar es Salaam and Mbeya regions in Tanzania [6]. Thus, n = 306 for each group, giving a total sample size of 612.

## Data collection

We interviewed participants to obtain socio-demographic characteristics including sex, age, marital status, occupation, and educational level; non-communicable disease risk factors such as smoking status and alcohol consumption; as well as the past medical history of hypertension and diabetes mellitus. Participants' CTC cards were reviewed to obtain information on ART status, type of ART, and duration on ART. The level of physical activity was assessed using the Short Last 7 days Self-administered Format of the International Physical Activity Questionnaire [7].

Observing the procedure for office blood pressure measurement outlined in the European Society of Hypertension/European Society of Cardiology Guidelines for the management of arterial hypertension, we measured blood pressure (BP) using a standardized digital BP machine (AD Medical Inc.), having seated a participant comfortably for 5 minutes. In a seating patient, measurements were made in both arms; the 1st and 5th Korotkoff sounds were used to identify the systolic and diastolic blood pressures, respectively. Two BP readings spaced 1–2 min apart and an additional measurement if the first two were quite different was taken; the average of the last two BP readings was recorded, and the measurement from the arm with higher BP considered for analysis [8].

In a patient wearing no shoes, we measured height with a stadiometer and recorded to the nearest 0.5 centimeters, and weight using a SECA weighing scale, recording to the nearest 0.5 kilograms. Body mass index was then computed, the interpretation of which was adapted from WHO [9].

Individuals who had been in a fasting state for the past ≥8 hours had fasting blood glucose (FBG) and laboratory investigations done on the same day of enrollment. Participants not in a fasting state were requested to come in a fasting state the next day for FBG and other laboratory investigations. A capillary fingertip blood sample was obtained from each patient for FBG determination and two-hour postprandial blood glucose was assessed for participants with impaired FBG after giving 75mg of oral glucose mixed with 200mls of water. GlucoPlus™ machines and check strips were used for blood glucose determination. Diabetes mellitus was

regarded as fasting plasma glucose (FPG) level ≥7 mmol/L or a 2-h plasma glucose (PG) level ≥11.1 mmol/L during the postprandial blood glucose test or a previously diagnosed diabetes mellitus; impaired fasting glucose as FPG levels between 5.6 and 6.9 mmol/L; and impaired glucose tolerance as 2-h postprandial PG levels between 7.8 and 11.0 mmol/L, as per the American Diabetes Association diagnostic criteria [10].

A 5ml venous blood sample was drawn from each patient and put in a red-toped vacutainer. Samples were stored in a cool box and transported to the MUHAS Clinical Research laboratory at the end of the day where biochemical tests were done on the same day using the COBAS INTEGRA® 400 plus analyzer (Roche Diagnostics), observing standard operating procedures. Factors analyzed included serum creatinine, total cholesterol, triglycerides, HDL cholesterol, and LDL cholesterol. Serum creatinine was used to estimate the glomerular filtration rate using the MDRD equation [11] and values <60ml/min/1.73m$^2$ were regarded as renal dysfunction [12].

## Statistical methods

Data was entered into EpiData version 3.1; IBM® SPSS® Statistics version 26 and Stata® version 13.0 were used for analysis. Categorical variables were summarized into frequencies and proportions. Differences in proportions across groups were compared using the Chi-square test. Continuous variables were summarized into means and standard deviations and compared between individuals with HIV infection on long-term ART and ART naïve using the student's t-test. We employed Poisson regression with robust standard errors to identify the association between variables and the diagnosis of hypertension, impaired glucose tolerance, diabetes mellitus, dyslipidemia, and renal dysfunction. Factors with p<0.2 in bivariate analysis were included in the multivariable analysis model. P-values <0.05 were considered statistically significant.

## Results

A total of 612 individuals were enrolled, half of whom were on ART for ≥5 years, referred to as long-term ART (LTART), and the other half were ART naïve.

## Socio-demographic and clinical characteristics of study participants

Table 1 summarizes the socio-demographic and clinical characteristics of the 612 individuals with HIV infection by ART status. Overall, female participants were 428 (69.9%), and by ART status, there were 220/306 (71.9%) and 208/306 (68%) in the LTART and ART naïve study groups respectively (p = 0.209). Participants aged ≥40 years were 74.8% (229/306) of those on LTART compared to 35.9% (110/306) among ART naïve subjects, p<0.001. There was no significant difference with regard to smoking in the two groups. Low-intensity physical activities were recorded in 94.1% (288/306) versus 78.8% (241/306) among ART naïve and those on LTART, respectively (p<0.001). Current alcohol use and being overweight or obese were more prevalent in LTART than ART-naive participants 33.0% (101/306) vs 22.5% (69/306), p = 0.004 and 57.2% (175/306) vs 28.4% (87/306), p<0.001, respectively (also see S1 Table).

## NCDs among study subjects

As depicted in Table 2, hypertension was found in 6.9% (21/306), 95% CI: 4.0–9.7%, of ART naïve and 25.2% (77/306), 95% CI: 20.3–30.0%, among LTART study subjects, p<0.001. Of the 306 ART naïve subjects, 14 (4.6%), 95% CI: 2.2–6.9, had impaired glucose tolerance compared to 22.9% (70/306), 95% CI: 18.2–27.6%, LTART study subjects, p<0.001. Likewise, 12 ART naïve subjects (3.9%), 95% CI: 1.7–6.1%, had diabetes mellitus compared to 52 (17.0%), 95% CI:

**Table 1. Comparison of socio-demographic and clinical characteristics of individuals with HIV infection on LTART with ART naïve.**

| Characteristic | Total N = 612 | ART status | | p-value |
|---|---|---|---|---|
| | | Naïve306 (50.0%) | ART ≥5 years306 (50.0%) | |
| **Mean age ± SD (years)** | 41.2 ± 12.8 | 37.5 ± 11.8 | 44.9 ± 12.7 | <0.001 |
| **Age groups (years)** | | | | |
| <40 | 273 (44.6%) | 196 (64.1%) | 77 (25.2%) | |
| ≥40 | 339 (55.4%) | 110 (35.9%) | 229 (74.8%) | <0.001 |
| **Sex** | | | | |
| Male | 184 (30.1%) | 98 (32.0%) | 86 (28.1%) | |
| Female | 428 (69.9%) | 208 (68.0%) | 220 (71.9%) | 0.290 |
| **Marital status** | | | | |
| Single | 174 (28.4%) | 98 (32.0%) | 76 (24.8%) | |
| Ever married | 438 (71.6%) | 208 (68.0%) | 230 (75.2%) | 0.049 |
| **Education level** | | | | |
| None | 46 (7.5%) | 33 (10.8%) | 13 (4.2%) | |
| Primary school | 358 (58.5%) | 203 (66.3%) | 155 (50.7%) | |
| Above primary school | 208 (34.0%) | 70 (22.9%) | 138 (45.1%) | <0.001 |
| **Occupation** | | | | |
| Not employed | 171 (27.9%) | 79 (25.8%) | 92 (30.1%) | |
| Employed | 115 (18.8%) | 37 (12.1%) | 78 (25.5%) | <0.001 |
| Self employed | 326 (53.3%) | 190 (62.1%) | 136 (44.4%) | |
| **Smoking** | | | | |
| No | 583 (95.3%) | 287 (93.8%) | 296 (96.7%) | |
| Yes | 29 (4.7%) | 19 (6.2%) | 10 (3.3%) | 0.087 |
| **Alcohol** | | | | |
| No | 442 (72.2%) | 237 (77.5%) | 205 (67.0%) | |
| Yes | 170 (27.8%) | 69 (22.5%) | 101 (33.0%) | 0.004 |
| **Level of physical activity** | | | | |
| Moderate /vigorous intensity | 83 (13.6%) | 18 (5.9%) | 65 (21.2%) | <0.001 |
| Low intensity | 529 (86.4%) | 288 (94.1%) | 241 (78.8%) | |
| **Mean BMI (kg/m$^2$)** | 24.8 ± 6.1 | 22.8 ± 4.8 | 26.9 ± 6.6 | <0.001 |
| **BMI** | | | | |
| Underweight/Normal | 350 (57.2%) | 219 (71.6%) | 131 (42.8%) | |
| Overweight/obesity | 262 (42.8) | 87 (28.4%) | 175 (57.2%) | <0.001 |
| **Family history of HTN** | | | | |
| No | 552 (90.2%) | 272 (88.9%) | 280 (91.5%) | |
| Yes | 60 (9.8%) | 34 (11.1%) | 26 (8.5%) | 0.277 |
| **Family history of DM** | | | | |
| No | 571 (93.3%) | 285 (93.1%) | 286 (93.5%) | |
| Yes | 41 (6.7%) | 21 (6.9%) | 20 (6.5%) | 0.872 |

ART: antiretroviral therapy; BMI: body mass index; DM: diabetes mellitus; HTN: hypertension.

12.8–21.2% subjects on LTART, p<0.001. Hypercholesterolemia existed in 51 (16.7%), 95% CI: 12.5–20.9% of ART naïve subjects compared to 93 (30.4%), 95% CI: 25.2–35.6%, among participants on LTART, p<0.001. Hypertriglyceridemia was found in 29 (9.5%), 95% CI: 6.2–12.8%, of ART naïve subjects compared to 16.0%, 95% CI: 11.9–20.1%, of subjects on LTART, p = 0.015. Renal dysfunction was observed in 30 (4.9%), 95% CI: 3.2–6.6%, participants and was similar between individuals on LTART and ART naïve (also see S2 Table).

**Table 2. Clinical and laboratory findings among 612 study subjects.**

| Variable | Total N = 612 | ART status | | p-value |
|---|---|---|---|---|
| | | ART naïve n = 306 | ART ≥5 years n = 306 | |
| **Hypertension** | | | | |
| Yes | 98 (16.0%) | 21 (6.9%) | 77 (25.2%) | <0.001 |
| No | 514 (84.0%) | 285 (93.1%) | 229 (74.8%) | |
| **Impaired glucose tolerance** | | | | |
| Yes | 84 (13.7%) | 14 (4.6%) | 70 (22.9%) | <0.001 |
| Other | 528 (86.3%) | 292 (95.4%) | 236 (77.1%) | |
| **Diabetes mellitus** | | | | |
| Yes | 64 (10.5%) | 12 (3.9%) | 52 (17.0%) | <0.001 |
| No | 548 (89.5%) | 294 (96.1%) | 254 (83.0%) | |
| **Renal dysfunction** | | | | |
| Yes | 30 (4.9%) | 17 (5.6%) | 13 (4.2%) | 0.454 |
| No | 582 (95.1%) | 289 (94.4%) | 293 (95.8%) | |
| **Hypercholesterolemia** | | | | |
| Yes | 144 (23.5%) | 51 (16.7%) | 93 (30.4%) | <0.001 |
| No | 468 (76.5%) | 255 (83.3%) | 213 (69.6%) | |
| **Hypertriglyceridemia** | | | | |
| Yes | 78 (12.7%) | 29 (9.5%) | 49 (16.0%) | 0.015 |
| No | 534 (87.3%) | 277 (90.5%) | 257 (84.0%) | |
| **Low HDL cholesterol** | | | | |
| Yes | 158 (25.8%) | 87 (28.4%) | 71 (23.2%) | 0.139 |
| No | 454 (74.2%) | 219 (71.6%) | 235 (76.8%) | |
| **High LDL cholesterol** | | | | |
| Yes | 157 (25.7%) | 87 (28.4%) | 70 (22.9%) | 0.116 |
| No | 455 (74.3%) | 219 (71.6%) | 236 (77.1%) | |

ART: antiretroviral therapy; HDL: high density lipoprotein; LDL: low density lipoprotein.

## Association between established risk factors and NCDs among study subjects

Factors independently associated with hypertension, diabetes mellitus, impaired glucose tolerance, hypercholesterolemia, and hypertriglyceridemia among individuals with HIV infection were: use of ART for ≥5 years, adjusted PR 1.36, 95% CI: 1.13–1.65, p = 0.001; age ≥40 years, adjusted PR 2.04, 95% CI: 1.63–2.56, p<0.001; smoking, adjusted PR 0.45, 95% CI: 0.24–0.86, p = 0.016 and being overweight/obese, adjusted PR 1.52, 95% CI: 1.28–1.80, p<0.001; see Table 3 (also see S3 Table).

## Discussion

Females constituted over two-thirds of the study participants; there was no significant difference in the sex distribution between the two arms of the study. Female predominance has been a consistent finding in hospital-based HIV studies in Tanzania and elsewhere in Africa [6,13–16]. This may be explained by the higher HIV prevalence rates among females compared to males in the Tanzania general population and the fact that females have higher health-seeking behavior than males [17]. The mean age of the study participants on long-term ART (LTART) was higher than that of ART naïve participants. This observation is similar to reports from other African studies [6,15,16] and could partly be due to beneficial effects of ART use on life

**Table 3. Factors associated with NCDs among 612 individuals with HIV infection.**

| Variable | Total N = 612 | NCDs* n = 290 | PR (95% CI) | p-value | Adjusted PR (95% CI) | p-value |
|---|---|---|---|---|---|---|
| **ART status** | | | | | | |
| Naïve | 306 | 96 (31.4%) | 1 | | 1 | |
| ≥5 years | 306 | 194 (63.4%) | 2.02 (1.68–2.44) | <0.001 | 1.36 (1.13–1.65) | 0.001 |
| **Age group (years)** | | | | | | |
| <40 | 273 | 70 (25.6%) | 1 | | 1 | |
| ≥40 | 339 | 220 (64.9%) | 2.53 (2.04–3.14) | <0.001 | 2.04 (1.63–2.56) | <0.001 |
| **Sex** | | | | | | |
| Male | 184 | 89 (48.4%) | 1 | | | |
| Female | 428 | 201 (47.0%) | 0.97 (0.81–1.16) | 0.748 | | |
| **Smoking status** | | | | | | |
| Yes | 29 | 6 (20.7%) | 0.42 (0.21–0.87) | 0.019 | 0.45 (0.24–0.86) | 0.016 |
| No | 583 | 284 (48.7%) | 1 | | 1 | |
| **Alcohol use** | | | | | | |
| Yes | 170 | 88 (51.8%) | 1.13 (0.95–1.35) | 0.168 | 0.95 (0.81–1.10) | 0.494 |
| No | 442 | 202 (45.7%) | 1 | | 1 | |
| **Physical activity** | | | | | | |
| Low intensity | 529 | 250 (47.3%) | 1 | | | |
| Moderate/Vigorous | 83 | 40 (48.2%) | 1.02 (0.80–1.30) | 0.873 | | |
| **BMI** | | | | | | |
| Underweight/Normal | 350 | 118 (33.7%) | 1 | | | |
| Overweight/obesity | 262 | 172 (65.6%) | 1.95 (1.64–2.31) | <0.001 | 1.52 (1.28–1.80) | <0.001 |

ART: antiretroviral therapy; BMI: body mass index; CI: confidence interval; NCDs: non-communicable diseases; PR: prevalence ratio.

* NCDs considered: hypertension, impaired glucose tolerance, diabetes mellitus, hypercholesterolemia, and hypertriglyceridemia.

expectancy. Moderate/vigorous-intensity physical activities were significantly more prevalent among individuals on LTART than ART naïve, which may be because of implementing health advice regularly offered to individuals at Care and Treatment Clinics.

The prevalence of hypertension was significantly higher among individuals on LTART compared to ART naïve participants and is in-keeping with findings from other studies in Tanzania, Cameroon, and Italy [6,16,18,19]. Furthermore, the prevalence of hypertension was observed to increase with the duration of ART use. These findings are similar to those from a multicenter AIDS cohort study which demonstrated a link between the duration of ART and elevated blood pressure, suggesting that prolonged ART use was independently associated with the development of hypertension [20]. Indeed, higher prevalence rates of hypertension ranging from 44.4%-48.2% were reported from other studies [21–23]. The high prevalence of hypertension among adults with HIV infection on LTART is likely to be multifactorial. It is important to point out here that following primary infection by HIV, the body usually mounts a robust cellular and humoral immune response but rarely able to clear the virus. Consequently, a chronic state characterized by elevated levels of antigen-antibody immune complexes sets in. Inflammation is well recognized in the pathophysiology of hypertension [24]. Hypertension was also associated with being aged 40 years and above similar to findings from several other studies in Africa [6,16,25]. Loss of elasticity in arterial blood vessels occurs with increasing age and is an important factor in the development of high blood pressure [26].

The overall prevalence of impaired glucose tolerance (IGT) was 22.9% among subjects on long-term ART and 4.6% in ART naïve subjects (p<0.001). The estimated IGT prevalence

among adults in the general population in Tanzania is 9.5% [27]. Consequently, the results of this study not only show high prevalence rates of IGT among individuals with HIV infection on ART but also show IGT rates increase with the duration of ART use. A study done in South Africa reported a prevalence of IGT of 3–4 fold among participants with HIV infection on ART compared to ART naïve participants [28]. Another study done in Mwanza Tanzania reported a 3.5-fold higher prevalence of IGT among ART experienced participants compared to ART naïve [29]. These findings affirm a strong association between ART use and increased prevalence of IGT. The increased rate of IGT among individuals with HIV infection on ART could be due to a number of factors. Such factors include HIV stress-induced hyperglycemia, type of ART used, and several other factors. We are not able to determine if any of these individuals with elevated IGT will develop diabetes mellitus in the future. Longitudinal studies may provide answers as to what proportion, if any, of individuals with ART use-related IGT go on to develop clinical diabetes mellitus.

The prevalence of diabetes mellitus (DM) was higher among subjects on LTART compared to the ART naïve group and was about 4-fold higher than the estimated prevalence among adults in the general population [27]. In a study conducted from CTC clinics in Dar es Salaam and Mbeya, Kagaruki et al reported prevalence rate of DM similar to that in the general adult population [6]. However, part of limitations of their study was poor turn-up of participants for follow-up biochemical tests including fasting blood glucose. Furthermore, their finding differs from most published data. The high prevalence of DM in our study is similar to reports from other studies [30–33]. A multicenter AIDS cohort study reported a fourfold increase in the prevalence of DM in HAART-exposed subjects [34]. The high prevalence of DM among individuals with HIV infection on LTART is most certainly multifactorial. ARTs such as protease inhibitors contribute to insulin resistance via a direct effect on insulin-mediated glucose transport leading to dysglycemia [35]. Because of ART use, survival among people with HIV infection has improved for age-related underlying genetic, diet, and lifestyle risks to developing type 2 diabetes mellitus and other NCDs, as it would be in the general population. Hence, individuals with HIV infection on ART should be regularly screened for DM, in particular, those above the age of 40 years.

The prevalence of renal dysfunction in this study was similar to a study from Cameroon [36] and did not appear to be associated with duration of ART use. Higher rates of renal dysfunction have been reported from other studies [15,37,38]. Renal disease in individuals with HIV infection is not uncommon and may be due to several factors including direct kidney injury by the virus and associated immune responses and/or ARV drugs such as tenofovir disoproxil fumarate and several others [39]. It is likely that study subjects on LTART may have had HIV infection for longer periods than ART naïve subjects. Nonetheless, there was no statistically significant difference in the prevalence of renal dysfunction between the two groups. Since our findings do not suggest increased rates of renal dysfunction among individuals on LTART, these drugs may reduce and/or delay the occurrence of kidney damage. However, the burden of renal diseases in individuals with HIV infection is complicated by the fact that the reduction in the prevalence of HIV-associated nephropathy accomplished by the administration of ART is offset by comorbid disease in the aging HIV population as well as ART nephrotoxicity [40].

Hypercholesterolemia was found in about a third of study subjects on LTART compared to 16.7% among ART naïve participants similar to reports of other studies in sub-Saharan Africa [6,41,42]. In vitro studies have pointed out the likely direct atherogenic effects of some antiretroviral medications like ritonavir on the vascular endothelium, macrophages, and platelets through the promotion of cholesterol accumulation or a reduction in cholesterol efflux from macrophages, decrease in endothelial nitric oxide production and cytotoxicity to endothelial

cells [24]. Likewise, hypertriglyceridemia was higher among participants on LTART compared to ART naïve similar to a report of a systematic review and meta-analysis [43].

Overweight/obesity was more prevalent among participants on LTART compared to ART naïve study subjects and was higher than the reported prevalence of overweight/obesity in a previous study in Dar es Salaam [44]. In the Temprano trial, the prevalence of overweight/obesity among individuals with HIV infection increased from 27 to 32 percent after 24 months of initiation of ART [45]. Our study was conducted in an urban setting where overweight and obesity are on the rise. Likewise, many countries have reported increased prevalence of overweight and obesity in persons with HIV infection even before ART initiation, consistent with trends in the general population. Of note is that overweight/obesity in people with HIV infection may be a result of several factors. It may reflect increased food intake to prevent weight loss considered to be a principal feature of AIDS in many communities. Indeed, there is a misconception in some communities/cultures that a person whose weight is increasing does not have HIV infection. Overweight and obesity may also result from chronic inflammation in HIV which predisposes the adipose tissue to become metabolically active and a source of bioactive peptides which, via a complex interplay of factors, lead to the release of cytokines, interleukins, and leptin, a key pro-inflammatory adipokine associated with inflammation in the setting of obesity. Some markers of inflammation have been associated with greater gains in fat after initiation of ART [46].

The selected NCDs were associated with long-term use of ART, advancing age, and being overweight/obese. The negative association between NCDs and smoking was unexpected and is contrary to known associations [47–49]. We are unable to adequately account for this finding. However, it is worth noting that the prevalence of smoking among participants on LTART was 3.3% and 6.2% among ART-naive. Furthermore, only 0.9% of females and 13.6% of males in this study reported being smokers compared to 27.9% among adult males and 7.2% among females in Dar es Salaam [50]. Since the health information given regularly during clinic visits includes the negative impacts of smoking on health, some participants may have heeded to the advice, hence the small number of study subjects who reported being smokers. On the other hand, participants may have underreported smoking for fear of offending the healthcare providers who would have advised them not to smoke or due to other reasons. Indeed, in many African cultures and practices, women are not expected to smoke, and hence, due to these social pressures, underreporting is likely to be common.

The findings from this study show a strong association between long-term ART use and high prevalence rates of NCDs among residents of Dar es Salaam similar to observations reported from Europe, North America, and elsewhere.

## Study limitation

This study was not designed to determine which particular antiretroviral drugs or regimens were associated with the selected NCDs. Serum creatinine was used to assess kidney function; this can underestimate kidney damage.

## Conclusion

Hypertension, impaired glucose tolerance, diabetes mellitus, hypercholesterolemia, and hypertriglyceridemia were significantly associated with long-term use of antiretroviral therapy. Individuals with HIV infection on long-term ART, especially those aged ≥40 years, require screening for NCDs.

## Supporting information

**S1 Table. Comparison of socio-demographic and clinical characteristics of individuals with HIV infection in 3 ART status groups.**
(DOCX)

**S2 Table. Clinical and laboratory findings among study subjects in 3 ART status groups.**
(DOCX)

**S3 Table. Factors associated with hypertension, impaired glucose tolerance, diabetes mellitus, renal dysfunction, hypercholesterolemia, hypertriglyceridemia, low HDL cholesterol, and high LDL cholesterol among individuals with HIV infection.**
(DOCX)

**S1 Appendix. Interview tool.**
(DOCX)

## Acknowledgments

The authors are grateful to Dr. Candida Moshiro for her guidance in statistical work and the research assistants who assisted in recruitment of study participants and data collection.

## Author Contributions

**Conceptualization:** Irene Kato, Kisali Pallangyo.

**Data curation:** Irene Kato.

**Formal analysis:** Irene Kato, Basil Tumaini, Kisali Pallangyo.

**Investigation:** Irene Kato.

**Methodology:** Irene Kato, Basil Tumaini, Kisali Pallangyo.

**Supervision:** Kisali Pallangyo.

**Writing – original draft:** Irene Kato, Basil Tumaini, Kisali Pallangyo.

**Writing – review & editing:** Irene Kato, Basil Tumaini, Kisali Pallangyo.

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
