## [Decision Letter · Decision Letter 0]

20 Feb 2020

PONE-D-20-00114

Prevalence of non-communicable diseases among HIV-infected patients by antiretroviral therapy status in Dar es Salaam, Tanzania

PLOS ONE

Dear Dr Kato,

Thank you for submitting your manuscript to PLOS ONE. After careful consideration, we feel that it has merit but does not fully meet PLOS ONE’s publication criteria as it currently stands. Therefore, we invite you to submit a revised version of the manuscript that addresses the points raised during the review process.

We would appreciate receiving your revised manuscript by 30th April 2020. To enhance the reproducibility of your results, we recommend that if applicable you deposit your laboratory protocols in protocols.io, where a protocol can be assigned its own identifier (DOI) such that it can be cited independently in the future. For instructions see: http://journals.plos.org/plosone/s/submission-guidelines#loc-laboratory-protocols

We look forward to receiving your revised manuscript.

Kind regards,

Kwasi Torpey, MD PhD MPH

Academic Editor

PLOS ONE

Journal Requirements:

2. Please include copies of the interview guide(s) used in the study, in both the original language and English, as Supporting Information, or include a citation if they have been published previously.

Reviewers' comments:

Reviewer's Responses to Questions

**Comments to the Author**

1. Is the manuscript technically sound, and do the data support the conclusions?

Reviewer #1: Yes

Reviewer #2: Partly

Reviewer #3: Partly

2. Has the statistical analysis been performed appropriately and rigorously? 

Reviewer #1: Yes

Reviewer #2: No

Reviewer #3: No

3. Have the authors made all data underlying the findings in their manuscript fully available?

Reviewer #1: Yes

Reviewer #2: Yes

Reviewer #3: Yes

4. Is the manuscript presented in an intelligible fashion and written in standard English?

Reviewer #1: Yes

Reviewer #2: Yes

Reviewer #3: Yes

5. Review Comments to the Author

Reviewer #1: Nicely written article. Although my primary research area is epidemiology of respiratory infections (with a focus on influenza/EID), the article was easy to read and understand. My concerns are highlighted below. Would also request you to run a plagiarism check in the discussion section for improving certain sentences.

• Research assistants are mentioned in methods. May include them in acknowledgements as appropriate.

• The study tool / questionnaire might be provided in annexure if appropriate. Specifically, I wanted to know if family history of HIV was enquired.

• Table 1 shows significantly higher number of people in ART 5+ years for “employed” and “moderate/vigorous intensity”. You might mention this in discussion with possible reasons if any (ex: specific employment/activity program by government/hospital/NGO)

• Discussion section: line nos 207, 226, 240, 249, 255-56, 261-62 might require some language editing.

• Line 239-241 refers to study from USA but the study listed in reference is from South Africa (reference number 39). Kindly check

Reviewer #2: PONE-D-20-00114

General comments

1. The study is well written but still requires some copy editing.

2. The authors should consider writing “participants with HIV” rather than HIV/AIDS patients. The people in the research are participants in the research rather than patients being treated. In addition, it is not good practice to put the disease (HIV) before the person (patient). They are people with the disease, not the disease with the people.

3. The authors should also consider removing the “AIDS” on HIV/AIDS, as the term AIDS usually implies advanced HIV infection.

Materials and methods

1. Were the questionnaires assessing smoking and alcohol validated, and how?

2. How many times were blood pressure and anthropometry measured per participant. How reliable were these measurements?

3. On line 101, the authors state, “HIV-infected patients who had been in a fasting state for the past 8 hours had fasting blood glucose and laboratory investigations done on the same day of enrolment”. How was fasting status ascertained?

4. How accurate is the GlucoPlus machine that was used for blood glucose measurement?

5. Please explain, briefly, what the ADA diagnostic criteria were.

6. On line 110, the authors state, “A 5ml venous blood sample was drawn from each patient and put in a red-toped vacutainer…”. Were these samples in the fasted state? What do the authors mean by “cool environment”, bearing in mind the humidity and high temperatures in Tanzania during the study period September to December.

7. Sample size – can the authors explain how the sample size was reached?

8. Exposures – the authors surely have data on the type of ART… Were all patients on one regimen, please describe the regimen and composition. If the regimen were different, perhaps an analysis based on these different regimen would be informative?

9. How was physical activity measured and then categorised?

10. Outcomes – the authors need to explain how each outcome was measured and defined and what guidelines and criteria were used.

11. On line 119, do the authors mean that all variables collected were categorical in nature?

12. Still on the analysis, what variables were considered for the logistic regression? What criteria was used for selecting these variables?

13. The authors state that there was a logistic regression, but do not explain how the outcome was defined.

14. The equal numbers between the groups are interesting. Was enrolment stopped when the numbers were equal?

15. In Results, lines 153-159, did the authors investigate whether this observed trend was not due to age?

16. Table 3 – logistic regression. Please explain why smoking is protective in this study, contrary to known effect.

17. In the logistic regression, continuous exposures should not have been categorised, unless there were strong reasons to do so, i.e. if the study aim was to investigate the effect of age>40 or BMI>certain cut-offs. The primary aim was to examine the effect of ART

18. The analysis requires re-working and the Discussion modified after that.

References

Reference number 1 “Joint United Nations Programme on HIV/AIDS (UNAIDS). Global Aids Update 2016. Geneva: UNAIDS, 2016.” is incomplete. Please add a link (URL) for where the resource can be accessed.

Reviewer #3: This is an important study in a setting where NCDs are becoming a priority research area. The aims were to determine the prevalence of NCDs and their risk factors amongst HIV infected patients on ART, with specific reference to the timing of ART use (long term vs naive). This would imply an internal control group that would allow the authors to infer the risk of time on ART for development of NCDs. The design is cross -sectional. I have identified some major methodological issues for the authors to address, which i think could have affected both the prevalence estimates as well as the risk factor associations.

1. Please report the assumptions used in the sample size estimation. What was the minimum difference of clinical importance?

2. For the risk factor analysis it appears that all NCDs were combined into a composite outcome. This is not specified nor defined anywhere. Composite outcomes themselves can be problematic especially where the risk factors for one of the outcomes is not the same as for the other outcomes. If you are using a composite outcome you should justify that it is sound to combine the outcomes and that valid inferences on risk factors to the composite outcome can be made. Otherwise, consider using each outcome as a separate outcome for the risk factor analysis.

3. The sampling design seemed to be cluster sampling, with the primary sampling units as hospitals and the secondary units as patients. This has an effect on prevalence estimates since prevalence within a specific facility can be more similar than between facilities. The authors need to incorporate cluster sampling design effect into their sample size calculation and into the analysis of the results.

4. I would like to see 95% confidence intervals reported around all your prevalence estimates. It is not merely sufficient to report the estimate on its own without making an inference to the population from which the sample was drawn.

5. In the risk factor analysis, logistic regression was used. This is not appropriate for estimating relative risks from a cross sectional study where the outcome is not rare (rare is <10%). Please see https://www.tandfonline.com/doi/full/10.1080/02664763.2013.840772 for more information. You could use a log-binomial model and it can be done in SPSS (https://www.ibm.com/support/pages/log-binomial-model) although I have never used it in this software and Stata or R would probably be a better option. Please consult a biostatistican to ensure you use the correct model and interpret it correctly. Remember also to adjust for the within-facility clustering due to the cluster sampling used.

6. In the discussion, the risk factors are not discussed at all.

6. PLOS authors have the option to publish the peer review history of their article (what does this mean?). If published, this will include your full peer review and any attached files.

Reviewer #1: No

Reviewer #2: Yes: Tawanda Chivese

Reviewer #3: Yes: Tonya Marianne Esterhuizen

---

## [Author Response · Author response to Decision Letter 0]

1 May 2020

Dear Editor,

We appreciate the opportunity to revise and resubmit our manuscript titled “Prevalence of non-communicable diseases among individuals with HIV infection by antiretroviral therapy status in Dar es Salaam, Tanzania”. We thank the Editor and reviewers for revision recommendations. We believe the revised manuscript is strengthened by the recommendations. We have provided a table with the reviewers’ comments and our responses. Two versions are submitted, one with tract changes and the other without as recommended.

 

Editor

Review comments Responses Location of the response in the revised manuscript

1. Please ensure that your manuscript meets PLOS ONE's style requirements, including those for file naming. File naming and text formatting for the manuscript and supporting information now meets PLOS ONE's requirements. -

2. Please include copies of the interview guide(s) used in the study, in both the original language and English, as Supporting Information, or include a citation if they have been published previously. A copy of the interview guide used in the study has been provided as Supporting Information. The investigator and research assistants used the English version and could translate it during the interview. S1 Appendix

 

Responses to Reviewers’ Comments

Reviewer 1

Review comments Responses Location of the response in the revised manuscript

Nicely written article. Although my primary research area is epidemiology of respiratory infections (with a focus on influenza/EID), the article was easy to read and understand. My concerns are highlighted below. Would also request you to run a plagiarism check in the discussion section for improving certain sentences. Thank you for the complements.

We have run a plagiarism check and made appropriate editing in the discussion. In the discussion

Research assistants are mentioned in methods. May include them in acknowledgements as appropriate. Research assistants are now included in the Acknowledgments instead of Materials and Methods. Line 102, lines 327-8

The study tool / questionnaire might be provided in annexure if appropriate. Specifically, I wanted to know if family history of HIV was enquired. We did not collect information about the family history of HIV infection.

The study tool is provided. S1 Appendix, attached

Table 1 shows significantly higher number of people in ART 5+ years for “employed” and “moderate/vigorous intensity”. You might mention this in discussion with possible reasons if any (ex: specific employment/activity program by government/hospital/NGO) The following description has been added in the discussion: “Moderate/vigorous-intensity physical activities were more prevalent among individuals on LTART than ART naïve, which may be because of implementing health advice regularly offered to individuals at Care and Treatment Clinics”. Lines 215-217

Discussion section: line nos 207, 226, 240, 249, 255-56, 261-62 might require some language editing. We have edited the discussion as suggested. In the discussion

Line 239-241 refers to study from USA but the study listed in reference is from South Africa (reference number 39). Kindly check Thank you for pointing out the error.

We have corrected the typographical error and rewritten the sentence. Lines 273-276

 

Reviewer 2

Review comments Responses Location of the response in the revised manuscript

General comments

1. The study is well written but still requires some copy editing. Thank you for the complement.

Copy editing has been undertaken. Throughout text

2. The authors should consider writing “participants with HIV” rather than HIV/AIDS patients. The people in the research are participants in the research rather than patients being treated. In addition, it is not good practice to put the disease (HIV) before the person (patient). They are people with the disease, not the disease with the people. Thank you for the advice.

Use of “HIV/AIDS patients” has been dropped from the manuscript and replaced with “individuals with HIV infection”. Throughout text

3. The authors should also consider removing the “AIDS” on HIV/AIDS, as the term AIDS usually implies advanced HIV infection. The word HIV disease has been used instead of HIV/AIDS, as recommended. Throughout text

Materials and methods

1. Were the questionnaires assessing smoking and alcohol validated, and how? We did not assess smoking and alcohol intake other than whether someone smokes or takes alcohol. Lines 102-104

2. How many times were blood pressure and anthropometry measured per participant. How reliable were these measurements? Blood pressure and anthropometric measurements were done at time of enrollment. The measurement of blood pressure was done according to the European Society of Hypertension/ European Society of Cardiology Guidelines for the management of arterial hypertension (Reference number 8 in the manuscript text). Lines 108-120

3. On line 101, the authors state, “HIV-infected patients who had been in a fasting state for the past 8 hours had fasting blood glucose and laboratory investigations done on the same day of enrolment”. How was fasting status ascertained? Individuals were asked about their last food intake. Those who had not taken food in the previous 8 hours or more were considered to be in a fasting state, and a blood sample was taken. Line 121-124

4. How accurate is the GlucoPlus machine that was used for blood glucose measurement? The machine is widely used, has been used in several studies, and is reliable. However, a study conducted in South Africa to assess the accuracy and reliability of glucose monitoring devices, including GlucoPlusTM reported satisfactory clinical accuracy, although all of the assessed devices had greater than 5% deviation (meeting ISO guidelines but missing ADA guidelines).

(Essack Y, Hoffman M, Rensburg M, Van Wyk J, Meyer CS, Erasmus R. A comparison of five glucometers in South Africa. Journal of Endocrinology, Metabolism and Diabetes of South Africa. 2009;14(2):102-5) Line 126-7

5. Please explain, briefly, what the ADA diagnostic criteria were. A brief description of the diagnostic criteria utilized has been included. Lines 127-132

6. On line 110, the authors state, “A 5ml venous blood sample was drawn from each patient and put in a red-toped vacutainer…”. Were these samples in the fasted state? What do the authors mean by “cool environment”, bearing in mind the humidity and high temperatures in Tanzania during the study period September to December. Blood samples were drawn on the first day from patients who had been fasting for at least 8 hours or next day for patients who were instructed to fast.

Samples were stored and transported to the laboratory in a cool box for biochemical analysis. “Cool environment” has been changed to “cool box” which is what was used. Lines 133-136

7. Sample size – can the authors explain how the sample size was reached? A section detailing the sample size calculation has been added in the Materials and Methods section. Lines 91-100

8. Exposures – the authors surely have data on the type of ART… Were all patients on one regimen, please describe the regimen and composition. If the regimen were different, perhaps an analysis based on these different regimen would be informative? Of 306 individuals with HIV infection on long-term ART, 174 (56.9%) reported having had a change of the ART regimen at least once. This may, therefore, complicate inferences on the association between ART regimens and NCDs, since information on the duration of different ART regimens was incomplete. -

9. How was physical activity measured and then categorised? Physical activity was assessed using the International Physical Activity Questionnaire – Short form (attached in Appendix 1). However, we included in the database and analyzed only whether a participant engages in vigorous physical activities (from question 1 & 2), moderate intensity physical activities (from question 3 & 4) or low intensity physical activities (question 5-7). Lines 106-7

S1 Appendix

10. Outcomes – the authors need to explain how each outcome was measured and defined and what guidelines and criteria were used. Guidelines and criteria used for each outcome are provided in the Materials and methods section of the manuscript. Line 108-110, 120, 127-132, 139

11. On line 119, do the authors mean that all variables collected were categorical in nature? We have edited the section to describe that both categorical and continuous variables were analyzed. Lines 142-6

12. Still on the analysis, what variables were considered for the logistic regression? What criteria was used for selecting these variables? The composite NCD included the following: hypertension, impaired glucose tolerance, diabetes mellitus, hypercholesterolemia, and hypertriglyceridemia. They are clinical manifestations or complications of metabolic syndrome. We aimed at determining the association between the above conditions and established risk factors for NCDs. Factors with p<0.2 in bivariate analysis were included in the multivariable Poisson regression with robust standard errors. Lines 146-150, 191-203

13. The authors state that there was a logistic regression, but do not explain how the outcome was defined. See our responses to comments number 10 and 12 above. (See also lines 202-203)

14. The equal numbers between the groups are interesting. Was enrolment stopped when the numbers were equal? Yes. The enrolment was stopped when the required number in a particular group was reached. Line 152-3

15. In Results, lines 153-159, did the authors investigate whether this observed trend was not due to age? Multivariable Poisson regression with robust standard errors (as in table 3) revealed that both ART status and age were independently associated with the NCDs. Lines 173-183

See also table 3 (line 197) and Supplementary Information, S3 Tables.

16. Table 3 – logistic regression. Please explain why smoking is protective in this study, contrary to known effect. This observation was in deep contrast to our expectations, and it is likely to be due to confounding factors. We have discussed the finding in the Discussion section. Lines 301-313

17. In the logistic regression, continuous exposures should not have been categorised, unless there were strong reasons to do so, i.e. if the study aim was to investigate the effect of age>40 or BMI>certain cut-offs. The primary aim was to examine the effect of ART Thank you for the suggestion. A table with a Poisson regression model without categorization of age and BMI is provided in the Supplementary information. We are, however, of the opinion that such categorization, e.g., of BMI, is of clinical importance and was thus presented in the main text. Line 191-199 (Table 3)

Supplementary Information (S3 Tables)

18. The analysis requires re-working and the Discussion modified after that. We did re-analysis and modified the respective sections. Lines 173-183, 191-199,

Table 3, S3 Tables

Discussion

References

Reference number 1 “Joint United Nations Programme on HIV/AIDS (UNAIDS). Global AIDS Update 2016. Geneva: UNAIDS, 2016.” is incomplete. Please add a link (URL) for where the resource can be accessed A URL has been added to reference number 1. Line 330-332

 

Reviewer 3

Review comments Responses Location of the response in the revised manuscript

This is an important study in a setting where NCDs are becoming a priority research area. The aims were to determine the prevalence of NCDs and their risk factors amongst HIV infected patients on ART, with specific reference to the timing of ART use (long term vs naive). This would imply an internal control group that would allow the authors to infer the risk of time on ART for development of NCDs. The design is cross -sectional. I have identified some major methodological issues for the authors to address, which i think could have affected both the prevalence estimates as well as the risk factor associations. Thank you for pointing out the significance of the study.

As stated in the last sentence of the Introduction, the study aimed at “determining the prevalence of selected NCDs and associated factors among individuals with HIV infection …” 

Our study was not designed to determine risk factors for NCDs among people with HIV infection. We are sorry for the misunderstanding. Lines 70-72

1. Please report the assumptions used in the sample size estimation. What was the minimum difference of clinical importance? A section on sample size estimation and assumptions made has been added. Lines 92-100

2. For the risk factor analysis it appears that all NCDs were combined into a composite outcome. This is not specified nor defined anywhere. Composite outcomes themselves can be problematic especially where the risk factors for one of the outcomes is not the same as for the other outcomes. If you are using a composite outcome you should justify that it is sound to combine the outcomes and that valid inferences on risk factors to the composite outcome can be made. Otherwise, consider using each outcome as a separate outcome for the risk factor analysis. We thank the reviewer for the comment.

Current analysis of NCD-associated factors considers the following into the composite outcome: hypertension, impaired glucose tolerance, diabetes mellitus, hypercholesterolemia, and hypertriglyceridemia (stated in text and footnotes of Table 3). The justification is that; they represent clinical manifestations or complications of metabolic syndrome. Indeed, the analysis of factors associated with each of the included outcomes individually yielded similar results. We have provided the analysis of each outcome in the supplementary information Lines 191-203 (Table 3)

Supplementary Information (S3 Tables)

3. The sampling design seemed to be cluster sampling, with the primary sampling units as hospitals and the secondary units as patients. This has an effect on prevalence estimates since prevalence within a specific facility can be more similar than between facilities. The authors need to incorporate cluster sampling design effect into their sample size calculation and into the analysis of the results. We did not employ cluster sampling. Although we included four different health care facilities, which tends to convey the idea of cluster sampling, the facilities receive individuals with HIV infection from all over Dar es Salaam. Since the facilities serve individuals from overlapping geographical areas in Dar es Salaam region and utilize the same Tanzania HIV treatment guidelines, we decided not to utilize cluster sampling design effect into sample size estimation and data analysis. There were over 80 administrative areas/wards included making analysis based on the specific geographical location not useful. Lines 81-90, 92-100, 146-8.

4. I would like to see 95% confidence intervals reported around all your prevalence estimates. It is not merely sufficient to report the estimate on its own without making an inference to the population from which the sample was drawn. We have added 95% confidence intervals around the prevalence estimates. Lines 173-183

5. In the risk factor analysis, logistic regression was used. This is not appropriate for estimating relative risks from a cross sectional study where the outcome is not rare (rare is <10%). Please see https://www.tandfonline.com/doi/full/10.1080/02664763.2013.840772 for more information. You could use a log-binomial model and it can be done in SPSS (https://www.ibm.com/support/pages/log-binomial-model) although I have never used it in this software and Stata or R would probably be a better option. Please consult a biostatistician to ensure you use the correct model and interpret it correctly. Remember also to adjust for the within-facility clustering due to the cluster sampling used. In assessing the variables for association with the NCDs, we present prevalence ratios and adjusted prevalence ratios instead of odds ratios and adjusted odds ratios. We used Poisson regression modeling with Log link and robust standard errors in Stata®. We have not adjusted for cluster design effect as that is not the sampling method utilized (as described in #3 above). Lines 191-201

(Also see lines 146-148)

6. In the discussion, the risk factors are not discussed at all. Thank you for the comment.

We have added a discussion on factors associated with the NCDs. Lines 301-313 and elsewhere in the Discussion

---

## [Decision Letter · Decision Letter 1]

10 Jun 2020

PONE-D-20-00114R1

Prevalence of non-communicable diseases among individuals with HIV infection by antiretroviral therapy status in Dar es Salaam, Tanzania

PLOS ONE

Dear Dr. Irene Kato,

Thank you for submitting your manuscript to PLOS ONE. After careful consideration, we feel that it has merit but does not fully meet PLOS ONE’s publication criteria as it currently stands. Therefore, we invite you to submit a r

We look forward to receiving your revised manuscript.

Kind regards,

Professor Kwasi Torpey, MD PhD MPH

Academic Editor

PLOS ONE

Additional Editor Comments (if provided):

A few comments are pending

Reviewers' comments:

Reviewer's Responses to Questions

**Comments to the Author**

1. If the authors have adequately addressed your comments raised in a previous round of review and you feel that this manuscript is now acceptable for publication, you may indicate that here to bypass the “Comments to the Author” section, enter your conflict of interest statement in the “Confidential to Editor” section, and submit your "Accept" recommendation.

Reviewer #2: (No Response)

Reviewer #3: All comments have been addressed

2. Is the manuscript technically sound, and do the data support the conclusions?

Reviewer #2: Yes

Reviewer #3: (No Response)

3. Has the statistical analysis been performed appropriately and rigorously? 

Reviewer #2: No

Reviewer #3: (No Response)

4. Have the authors made all data underlying the findings in their manuscript fully available?

Reviewer #2: Yes

Reviewer #3: (No Response)

5. Is the manuscript presented in an intelligible fashion and written in standard English?

Reviewer #2: Yes

Reviewer #3: (No Response)

6. Review Comments to the Author

Reviewer #2: General comments

The authors have addressed most of the initial suggestions very well. The statistical analysis still needs a little bit more input though:

1. Is there any justification for the statistical approach to modelling used here? In general p-values are not advisable as the criteria for inclusion of variables into models especially in observational studies where p-values could be significant due to chance and/or confounding. Variables that have either biological plausibility and/or evidence of association with the outcome could be included in models despite p-values. In this case, it appears that the variables included have both biological plausibility and literature-based associations with the outcome, so the authors need not change the analysis.

2. The protective effect of smoking reported in this study is, as the authors rightly point out, most likely due to measurement error. Further, the proportion of smokers (29/612 = 5%), and the numbers with outcome (n =6) are very small and may lead to biased estimates of the association. Smoking was not assessed with a validated questionnaire and underreporting is a big concern here. My humble suggestion is for the authors to remove the variable and explain this in either statistical methods or discussion. This may have more benefit than leaving the finding in the table where it may be misinterpreted with the associated potential for harm. I suspect that the model will remain largely unchanged except for the removal of smoking.

Reviewer #3: I have reviewed the revised manuscript based on my and other reviewer’s suggestions. It is now ready for publication and I have no further changes which need to be made.

1. Recommendation: Accept

7. PLOS authors have the option to publish the peer review history of their article (what does this mean?). If published, this will include your full peer review and any attached files.

Reviewer #2: Yes: Tawanda Chivese

Reviewer #3: Yes: Tonya Marianne Esterhuizen

---

## [Author Response · Author response to Decision Letter 1]

16 Jun 2020

Dear Editor,

We appreciate the opportunity to revise and resubmit our manuscript titled “Prevalence of non-communicable diseases among individuals with HIV infection by antiretroviral therapy status in Dar es Salaam, Tanzania”. We thank the Editor and reviewers for revision recommendations. We believe the revised manuscript has been strengthened by the recommendations. We provide a response to the reviewer’s comments, a manuscript version with tract changes, and the other without as recommended.

General comments 

The authors have addressed most of the initial suggestions very well. The statistical analysis still needs a little bit more input though:

1. Is there any justification for the statistical approach to modelling used here? In general p-values are not advisable as the criteria for inclusion of variables into models especially in observational studies where p-values could be significant due to chance and/or confounding. Variables that have either biological plausibility and/or evidence of association with the outcome could be included in models despite p-values. In this case, it appears that the variables included have both biological plausibility and literature-based associations with the outcome, so the authors need not change the analysis.

Response:

We note the comments from the reviewer and agree that the analysis remains as presented.

2. The protective effect of smoking reported in this study is, as the authors rightly point out, most likely due to measurement error. Further, the proportion of smokers (29/612 = 5%), and the numbers with outcome (n =6) are very small and may lead to biased estimates of the association. Smoking was not assessed with a validated questionnaire and underreporting is a big concern here. My humble suggestion is for the authors to remove the variable and explain this in either statistical methods or discussion. This may have more benefit than leaving the finding in the table where it may be misinterpreted with the associated potential for harm. I suspect that the model will remain largely unchanged except for the removal of smoking.

Response:

Smoking is a known important factor associated with the development and/or severity of NCDs and should be elicited for in a study on the subject. 

We share the views of the reviewer that the apparent ‘protective effect of smoking on NCDs’ as shown in the results section, could be misinterpreted and it is important to prevent this from happening. However, we are uncomfortable with the suggestion that observations on smoking be ‘swept under the carpet’. We believe that we are expected and indeed obliged, to explain the unexpected observation that ‘Smoking is protective to NCDs’. Indeed, we believe most of your readers would want to find out reasons for the observation against the universally accepted norm, which is what we have done in the discussion (interpretation of the finding/result). Consequently, we believe that data on smoking should be presented in the results section. However, having presented our reasoning, we would accept the decision of the editor on the matter.

---

## [Editor Report · Decision Letter 2]

18 Jun 2020

Prevalence of non-communicable diseases among individuals with HIV infection by antiretroviral therapy status in Dar es Salaam, Tanzania

PONE-D-20-00114R2

Dear Dr. Kato,

We’re pleased to inform you that your manuscript has been judged scientifically suitable for publication and will be formally accepted for publication once it meets all outstanding technical requirements.

Kind regards,

Professor Kwasi Torpey, MD PhD MPH

Academic Editor

PLOS ONE
---

## [Editor Report · Acceptance letter]

26 Jun 2020

PONE-D-20-00114R2 

Prevalence of non-communicable diseases among individuals with HIV infection by antiretroviral therapy status in Dar es Salaam, Tanzania 

Dear Dr. Kato:

I'm pleased to inform you that your manuscript has been deemed suitable for publication in PLOS ONE. Congratulations! Your manuscript is now with our production department. 

Kind regards, 

on behalf of

Professor Kwasi Torpey 

Academic Editor

PLOS ONE